# Integration of Square Fiducial Markers in Patient-Specific Instrumentation and Their Applicability in Knee Surgery

**DOI:** 10.3390/jpm13050727

**Published:** 2023-04-25

**Authors:** Vicente J. León-Muñoz, Joaquín Moya-Angeler, Mirian López-López, Alonso J. Lisón-Almagro, Francisco Martínez-Martínez, Fernando Santonja-Medina

**Affiliations:** 1Department of Orthopaedic Surgery and Traumatology, Hospital General Universitario Reina Sofía, 30003 Murcia, Spain; jmoyaangeler@gmail.com (J.M.-A.); drajla78@gmail.com (A.J.L.-A.); 2Instituto de Cirugía Avanzada de la Rodilla (ICAR), 30005 Murcia, Spain; 3Subdirección General de Tecnologías de la Información, Servicio Murciano de Salud, 30100 Murcia, Spain; mirindalopez@gmail.com; 4Department of Orthopaedic Surgery and Traumatology, Hospital Clínico Universitario Virgen de la Arrixaca, 30120 Murcia, Spain; fmtnez@gmail.com (F.M.-M.); fernando@santonjatrauma.es (F.S.-M.); 5Department of Surgery, Pediatrics and Obstetrics & Gynecology, Faculty of Medicine, University of Murcia, 30120 Murcia, Spain

**Keywords:** augmented reality (AR), virtual reality (VR), extended reality (XR), patient-specific instrumentation (PSI), surgical navigation system, immersive technology, knee

## Abstract

Computer technologies play a crucial role in orthopaedic surgery and are essential in personalising different treatments. Recent advances allow the usage of augmented reality (AR) for many orthopaedic procedures, which include different types of knee surgery. AR assigns the interaction between virtual environments and the physical world, allowing both to intermingle (AR superimposes information on real objects in real-time) through an optical device and allows personalising different processes for each patient. This article aims to describe the integration of fiducial markers in planning knee surgeries and to perform a narrative description of the latest publications on AR applications in knee surgery. Augmented reality-assisted knee surgery is an emerging set of techniques that can increase accuracy, efficiency, and safety and decrease the radiation exposure (in some surgical procedures, such as osteotomies) of other conventional methods. Initial clinical experience with AR projection based on ArUco-type artificial marker sensors has shown promising results and received positive operator feedback. Once initial clinical safety and efficacy have been demonstrated, the continued experience should be studied to validate this technology and generate further innovation in this rapidly evolving field.

## 1. Introduction

Nowadays, computer technologies play a crucial role in orthopaedic surgery and are essential in personalising many treatments. In the last two decades, assistive technologies have been progressively developed to increase knee arthroplasties’ accuracy and reproducibility [1,2]. In the late 1990s, computer-assisted surgery (CAS) started to gain interest [2,3]. CAS replaces the surgeon’s visual references with real-time monitoring of the execution of the technique by computerized devices, with the primary objective of increasing geometric precision. With CAS, a mean error statistically less than 1 mm for determining single points or distances and less than 1° for determining angles (*p* < 0.001) has been estimated [4]. More recently, accelerometer-based navigation systems, which use sterile single-use devices within the operative field, have been employed in knee arthroplasty surgery [5,6]. The rapid and continuous development of different medical imaging modalities (mainly computed tomography and magnetic resonance imaging) and advanced technologies for the processing of these images has allowed the combination of radiological techniques with various CAD (Computer Aided Design) tools and CAD/CAM (Computer Aided Manufacturing) processes [7]. This synergy allows the planning of knee replacement surgery on three-dimensional (3D) virtual models and the design and additive manufacturing of patient-specific instrumentation (PSI). The latest assistive technology gaining huge interest is robotically assisted surgery in knee arthroplasty [2]. In addition, changes in alignment paradigms are furthering the rationale for the need for robotics and artificial intelligence-based tools in prosthetic knee surgery [8].

Along with robotics, several authors are implementing augmented reality (AR) in different aspects of total knee arthroplasty (TKA) surgery [9,10,11,12,13,14,15]. New platforms now use AR as a critical differentiator to enhance the surgeon’s experience during surgery. The integration of virtual reality (VR) and AR allows live and virtual images to be obtained in the robot-assisted user interface, facilitating the surgeon’s ability to position and manipulate robotic instruments [16].

Nevertheless, knee surgery is not limited to prosthetic surgery. Innovations have indeed been incorporated later in non-replacement surgeries. Recent advances allow the usage of AR for many orthopaedic procedures [17,18,19,20,21,22]. To the best of our knowledge, the first to use the term AR was Thomas P. Caudell, a Boeing researcher who coined the term in 1990 [23]. In 1992, Caudell and Mizell developed a prototype that allowed a computer-produced diagram to be superimposed and stabilized on a specific position on a real-world object [24]. Milgram and Kishino described in 1994 the overlap between the physical and digital worlds and placed AR in this reality–virtuality continuum [25]. Extended reality (XR) is a combination of the so-called immersive technologies: VR (virtual reality) + AR (augmented reality) + MR (Mixed Reality). AR assigns the interaction between virtual environments and the physical world, allowing both to intermingle (AR superimposes information on real objects in real time) through an optical device (usually smart glasses). An alternative for AR surgery is optical surgical navigation with ArUco-type artificial marker sensors. ArUco is a minimal library for AR applications based exclusively on OpenCV that relies on b/w markers with codes detected by calling a single function [26]. OpenCV (Open Source Computer Vision Library) is an open-source (an Apache 2 licensed product) computer vision and machine learning software library written natively in C++. OpenCV was built to provide a common infrastructure for computer vision applications and to accelerate the use of machine perception [27]. Some potential benefits of integrating this technology into knee surgery are (1) improvement in accuracy in comparison to traditional surgical techniques, (2) gain of the surgeon’s attention to the surgical field, (3) reduction of possible exposure to harmful radiation, (4) significant reduction of procedure time and costs associated with surgery, (5) improvement in operating room efficiency, (6) educational usefulness, (7) and the creation of positive synergies between engineering and medicine that favour the development of bioengineering and the pursuit of surgical excellence [17,19,28,29,30].

This article aims to provide an overview of the use of projection-based AR (digital content superimposed on a live video feed of the real world displayed on a monitor) with ArUco-type artificial marker sensors in knee surgical techniques other than knee replacement surgery, highlight the benefits and challenges associated with their usage, and provide a narrative review of the existing literature on AR applications in knee surgery.

## 2. Description of the Process, Design, and Implementation of the Devices

The process starts with the digitized acquisition of computed tomography (CT) images of the knee (or of the hip, knee, and ankle in cases requiring surgical axis correction). For the acquisition, the patient must be supine in the centre of the gantry. The leg of interest for the study should be in full extension. The digitized images are stored in the Digital Imaging and Communications in Medicine (DICOM) standard. The scans should be in the same coordinate system (reference frame). Each acquisition should be accurately centred and magnified to ensure that the field of view (FOV) maximizes the region of interest. The FOV should be as small as possible if it fully images the joint. Ideally, it should be less than 200 mm, leading to a pixel dimension (defined as the ratio of the FOV to the acquisition matrix) of less than 0.39 mm for an acquisition matrix of 512 × 512 pixels. The recommendation is a slice thickness of less than 1 mm in the knee region (in cases requiring hips and ankles, the recommendation is to slice between 2 and 4 mm in these areas). The space between cuts should be, at most, the thickness of the cut. We use a tube voltage of 120 kV or higher and a current of 50 to 75 mA.

The bioengineers obtain the 3D model of each specific knee to be operated on with its anatomical characteristics and particularities (in the case of CT studies of the bone morphology of the knee) using segmentation. According to some authors, segmentation requires a trained operator, as no fully automated segmentation algorithms exist [7,31]. Other authors, on the other hand, have published remarkable accuracy of automatic segmentation by implementing a neural network architecture based on deep learning [32,33,34]. In our usual practice, this process is carried out by the bioengineers using Mimics^®^ software v.23 (Materialise, Leuven, Belgium) for the segmentation process. Mimics^®^ is an interactive tool for the visualization and segmentation of CT and Magnetic Resonance Imaging (MRI) images and 3D rendering of objects. The operator imports files in DICOM format into Mimics^®^ and selects from the series the images in which the bony structures are visualized with the least number of unnatural holes. To do so, he generates a mask for each series. The engineer selects the bone threshold and manually separates the different bone structures using the mask-splitting tool. The operator artificially selects each bone structure as a mask to be segmented, renames it, and assigns it a specific colour (to differentiate the structures and to be able to work with some of them independently from the others). The mask is then edited to reduce noise (image artifacts). The selected bone structures are converted to 3D models (software can output a virtual 3D reconstruction by algorithm analysis), and keeping the originals, they are optimized using different tools that smooth the model and reduce the number of triangles without altering the original bone geometry. Once finishing the process, all the information related to the 3D model is archived as an “.stl” file, the format commonly used for 3D printing and CAD.

For the planning process, the bioengineers of PQx Planificación Quirúrgica (PQx Planificación Quirúrgica, Murcia, Spain), a MedTech Start Up with which we plan our interventions, use the TopSolid’Design software (TopSolid, Évry, France). TopSolid is an integrated CAD/CAM software for designing and creating fully functional 3D parts. The bioengineer will plan the surgery with the different TopSolid tools on the 3D model according to the surgeon’s specifications. For knee surgery, especially for angular correction surgery of the limb, it is necessary to know the limb’s anatomy and mechanical particularities beforehand. The bioengineer draws the axes to determine the limb’s preoperative angular and planned values. These axes are obtained by defining the specific start and end points of the straight-line segments that define these axes and different reference points or coordinates of anatomical structures. Similarly to the way we use to obtain virtual models for planning replacement surgery [35,36], the bioengineer defines the centre of the hip as the centre of the sphere that delimits the femoral head. The centre of the distal femur is at the middle of the intercondylar notch, at the most distal point of the trochlear rim [35,36]. The straight-line segment between the point at the hip’s centre and the point at the middle of the intercondylar notch defines the femoral mechanical axis. The core of the proximal tibia is the midpoint of a line drawn between the intercondylar eminences (centre of the notch between the tibial spines), with an anterior displacement in the axial plane of 2 mm. The centre of the ankle or tibiotalar joint is a point at the middle of the line joining the most prominent part of the medial malleolus and the most prominent distal part of the lateral malleolus (the distal end or tip of the fibula). The straight-line segment between the point at the proximal tibial centre and the point at the core of the ankle defines the tibial mechanical axis. The angle formed in the coronal plane by the intersection of the femoral mechanical axis and the tibial mechanical axis describes the mechanical femorotibial angle or Hip-Knee-Ankle angle (HKA). In the virtual model, the operator identifies the most distal bony points of the medial and lateral condyles. The line joining these points (the tangent to the most distal ends of both condyles) defines the femoral articular axis. The angle formed in the coronal plane by the intersection of the femoral mechanical axis and the femoral joint axis defines the so-called lateral femoral distal angle or femoral mechanical angle determined on the lateral aspect. The deepest points of the centre of the medial and lateral tibial plates are defined. The line joining these points defines the tibial joint axis or joint line. The angle formed in the coronal plane on the medial aspect by the intersection of the mechanical tibial axis and the tibial joint axis defines the proximal or mechanical tibial angle. At the medial tibial plateau level, two reference points (one anterior and one posterior, excluding possible osteophytes) are determined to draw a line defining the tibial slope concerning the tibial mechanical axis in the sagittal plane. After determining the native angular characteristics of the limb, the engineer plans the intervention to be performed with the values proposed by the surgeon.

The PQx engineers use Python 3.8.3 (Python Software Foundation, Wilmington, NC, USA) as the programming language to develop the marker detection software and OpenCV 4.0.1 as the computer vision library. The engineers use Unity 2019.2.17f1 (Unity Software Inc., San Francisco, CA, USA) for the graphics engine. The method used to calculate distances is the one proposed by Vector3.Distance: the system receives two points, a and b, in three dimensions and performs the magnitude operation (a-b); the magnitude operation being √((ax−bx)^2+(ay−by)^2+(az−bz)^2).

The angular calculation method used is the one proposed by Vector3.Angle: the system receives two three-dimensional vectors representing the two directions whose degree difference is to be found. These vectors are normalized, and then the scalar product is performed; the result is restricted between −1 and 1, its arccosine obtained and multiplied by 57.29578 (180/pi) to transform radians to degrees returning the result in this magnitude.

Once the surgeon validates the planning, the ArUco markers, anatomical models, and cutting or tunnelling templates are produced using a 3D printing system [37]. The bioengineers at PQx currently use the Ultimaker S3 printers (Ultimaker BV, Utrecht, The Netherlands) with double extrusion and a print volume of 230 mm × 190 mm × 200 mm. The printing is done on Ultimaker polylactic acid (PLA) filament (Ultimaker BV, Geldermalsen, The Netherlands).

The planning and navigation system we have used combines virtual and AR. Planning and calculations are performed on the virtual model. The markers are needed as references to project the virtual planning onto AR. AR in isolation does not provide a basis in the three axes of space for precise measurements from point A (x, y, z) to point B (x, y, z).

During surgery, the optical camera will detect the markers and capture the images of the real world. The physical guide points are the reference to show the virtual world through the screen of our device. The processor is the element in charge of combining the images to form the AR through the real images and the information from the virtual world. A specific software controls the processor. The software is the computing or logical element that manages all the previous processes of the camera and the processor. The return in the form of an image will be a superimposition of the virtual over the real [38].

We have employed for our earliest surgeries the Logitech C920 Full HD (1080p at 30 fps) webcam (Logitech, Lausanne, Switzerland) with a diagonal field of view (dFoV) of 78° and automatic HD illumination correction. We have recently switched to OAK-D (A00110-INTL) cameras (Luxonis Holding Corporation, Denver, United States) for their improved features. The OAK-D baseboard has three on-board cameras which implement stereo and RGB vision, piped directly into the OAK system on modules for depth and Artificial Intelligence processing, with a dFoV of 81° and a resolution of 12MP (4032 × 3040).

To date, we have used this type of technology to perform a variety of knee surgeries: femoral and tibial osteotomies (for the correction of varus, valgus, or rotational deformities), primary and revision surgery of the anterior and posterior cruciate ligaments, and medial patellofemoral ligament reconstruction, meniscal transplants, and different combinations of these procedures. Figure 1, Figure 2 and Figure 3 show the planning of a tibial opening wedge osteotomy and targeted tunnelling for the posterior and anterior anchorage of medial meniscal transplantation for an incipient medial femorotibial osteoarthritis in a young patient with a previous meniscectomy. Figure 4 and Figure 5 show the planning of a de-rotational osteotomy to correct excessive femoral anteversion in a previous hip replacement revision surgery where femoral stem revision was not feasible.

## 3. A Narrative Review on AR Applications in Knee Surgery

We can divide the publications that refer to the application of AR in knee surgery into those related to replacement surgery, those related to learning based on immersive technologies, arthroscopic surgery assistance (such as ligament surgery), remote surgery assistance, techniques related to the rehabilitation process and gait analysis studies.

Most articles addressing AR’s usefulness in knee surgery refer to replacement surgery. For example, Daniel and Ramos presented 2016 the design of a TKA surgical assistance system, using MRI to build 3D models of the tibia and femur and AR to visualize the bone osteotomy to be performed and compared with the one planned on the model, highlighting the cost–benefit ratio of the implemented system [39]. Pokhrel et al. [40] proposed a new matching approach during the iteration of pair point matching to minimize the error metric in the iterative closest point (ICP) (algorithm employed to reduce the difference between two clouds of points). This results in a reduction of the cutting error by about 1 mm and improves the image processing time. Therefore, the accuracy parameters published with AR are similar to those published with CAS [4]. Tsukada et al. [15] presented a preclinical pilot study to evaluate the accuracy of coronal, sagittal, and rotational alignment in tibial bone resection during TKA assisted by an AR system. The authors reported results indicating that the evaluated AR system had comparable accuracy to conventional navigation systems in varus/valgus, posterior slope, and internal/external rotation angles [15]. Subsequently, Tsukada et al. [10] published the accuracy of distal femoral resection during TKA, with a pilot study on femoral SawBone specimens and a clinical study comparing the AR system with the conventional intramedullary guide. The authors reported a mean error in distal femoral resection of less than 0.1° in the coronal and sagittal planes with the AR system in the experimental setting and significantly higher accuracy during distal resection in TKA compared to conventional intramedullary guide in the clinical setting [10]. Another exciting aspect offered by AR is surgical visualization. For example, Wang et al. [41] published the usability of the Microsoft HoloLens-based AR navigation system for minimally invasive TKA with real-time intuitive surgical visualization, overlaying the virtual model in the field of view accurately via holographic space calibration and image registration. Other authors [42] also suggested that AR technology will undoubtedly play an essential role in assisting joint replacement surgery, improving the precision of implantation with better intraoperative ergonomics and workflow without adding high extra cost to the procedure. Another aspect of great interest in replacement surgery is the ability to obtain real-time information on the stable or unstable behaviour of the native knee and the prosthesis during surgery. In this respect, Fucentese et al. [14] described an innovative AR-based surgical guidance system that measures the effect of prosthesis alignment intraoperatively and positioning on soft tissue balance (NextAR TKA, Medacta International SA, Castel San Pietro, Switzerland). It is the first AR-based guidance system officially cleared for use in TKA. Thus, by knowing the ligamentous insertions in x-y-z coordinate axes from, for example, DICOM images of CT scans using compact infrared sensors, the surgeon can have control over distances and information on the elongation–tension relationship in the AR projection. Iacono et al. [11] published a systematic review of the literature and a pilot clinical study of the use of AR for limb and component alignment in TKA. The authors found only two studies [15,43] concerned TKA; unfortunately, both were preclinical studies. In our opinion, the study by Fallavollita et al. [43] is broader. It deals with the applicability of AR C-arm for intraoperative assessment of the mechanical axis in all knee surgeries that may be of interest (e.g., axis correction osteotomies). Iacono et al. [11] presented preliminary results using Knee+ AR navigation intraoperative assistance for implant positioning with the help of AR glasses (Pixee Medical Company, Besançon, France). The Knee+ system consists of smart glasses worn by the surgeon, a laptop, and specific markers connected with tibial and femur resection guides [11]. Iterative Closest Point (ICP) algorithms were introduced in the early 1990s to register 3D range data to CAD models of objects. Maharjan et al. [13] proposed an ICP algorithm with bidirectional maximum correntropy. According to the authors, this algorithm helped improve registration and alignment, maximize the overlapping parts between two cloud points and eliminate the registration outcomes trapped in local minima. Furthermore, it has also removed the non-Gaussian noise, impulse noise, and outliers and provided high registration accuracy, helping in accurate visualization and navigation of the knee anatomy [13]. Recently, Su et al. [44] published a report on MR technology in preoperative planning and intraoperative navigation guidance for TKA.

Immersive learning outside the operating room is increasingly recognized as a valuable method of surgical training [45]. It is changing the model of surgical education credited to Dr W. Halsted of Johns Hopkins University in the nineteenth century [46] (a dogmatic model which requires trainees to spend long hours in the hospital to acquire the knowledge and mastery of skills necessary to advance in their training and autonomy as a surgeon) [9]. Some authors have used XR technologies for training in replacement surgery. Thus, Edwards et al. [47] have shown that immersive VR training improves scrub nurses’ understanding, technical skills, and efficiency in complex revision knee arthroplasty surgery. Zaid et al. [48] carried out a randomized controlled trial of trainees at a single, large academic centre performing a relatively complex and unfamiliar procedure (unicompartmental knee replacement); VR training demonstrated equivalent SawBone model surgical competence compared with the use of traditional technique guides, as measured by surgical time and the Objective Structured Assessment of Technical Skills scores. The authors suggested that VR technology could be considered an adjunct to traditional surgical preparation/training methods.

Studies on the applicability of XR in non-replacement knee surgery are scarce. As aforementioned, Fallavollita et al. [43] study aims to evaluate a new AR technology for determining lower extremity alignment. The authors obtained a Pearson’s R correlation coefficient value of 0.979, demonstrating a strong positive correlation between the camera-augmented mobile C-arm (AR technology) and the ground-truth CT data. The result of the analysis of variance showed that the differences in clinician experience were not significant.

AR has begun to be used to improve the accuracy of knee ligament repair surgeries. Bone tunnel placement is critical in avoiding complications and improving functional outcomes in knee ligament surgery. Preoperative planning can improve tunnel positioning [49]. However, translating the preoperative plan into the surgical procedure can be challenging. Guo et al. [50] proposed a bone-characteristics-based 2D–3D registration method for an anterior cruciate ligament (ACL) reconstruction navigation system, which could provide better guidance for the accuracy of tunnel placement. The authors concluded that a combination of splatting (for digitally reconstructed radiograph generation), Spearman’s rank correlation coefficient (for measurement of similarity between two images), and gradient descent (in iterative optimization for finding the maximum similarity of two images), provided the best composite performance for the AR ACL reconstruction navigation system. The accuracy of the navigation system could fulfil the clinical needs of ACL reconstruction, with an end pose error of 2.5 mm and an angle error of 2.74° [50].

Chen et al. [51] described an AR navigation assistance for arthroscopic knee surgery. The authors proposed a model deformation method based on tissue properties to update the preoperative 3D tissue structure according to the intraoperative arthroscopic view. In addition, they generated virtual arthroscopic images from the updated preoperative model to provide the anatomical information of the operation site [51].

AR has also been used to enable the interconnectivity between different actors in distant locations in the same surgery. Thus, van der Putten et al. [52] demonstrated the feasibility of providing remote telesurgical support with the HoloLens 2 head-mounted lens and the Microsoft Dynamics 365 Remote Assist software (Microsoft, Redmond, Washington, DC, USA) in TKA surgery. A proof-of-concept team consisting of an orthopaedic surgeon, the manufacturer’s account manager, and a proof-of-concept lead assisted the surgical team. Through MR, they solved a TKA surgery that required an implant with which the surgical team was unfamiliar, with a real-time resolution of the step-by-step intervention and the difficulties encountered.

Another area related to knee surgery in which the potential usefulness of immersive technologies has been published is rehabilitation. Thus, for example, Blasco et al. [53] published a systematic review that included six trials to assess the effects of training with virtual reality tools during the rehabilitation of patients after knee surgery. The authors highlighted the usefulness of improving balance and the lack of advantages over conventional rehabilitation in improving function, resolving pain, or increasing patient satisfaction after surgery [53]. Fung et al. [54] determined in a preliminary randomized controlled trial whether Nintendo Wii Fit™ (Nintendo of America Inc, Redmond, Washington, DC, USA) is an acceptable adjunct to physiotherapy treatment in the rehabilitation of balance, lower extremity movement, strength and function in outpatients following TKA. Nintendo Wii Fit™ includes a balance board similar in concept to a force plate. There were no significant differences in pain, knee range of motion, walking speed, timed standing tasks, self-perceived balance confidence, and self-perceived lower extremity function between the groups. Piqueras et al. [55] compared with a single-blind, randomized, controlled non-inferiority trial the effectiveness of a new interactive virtual telerehabilitation system and a conventional program following TKA. The virtual telerehabilitation system consists of interactive software with a 3D avatar demonstrating the exercises to be undertaken, sensors (3-axis accelerometer and gyroscopes) connected to the patient allowing calculation of their movement trajectories, and a web portal for the therapist. The physiotherapist receives data and records them for evaluation, with the option to modify the therapy as the rehabilitation evolves. The authors concluded that the use of interactive virtual telerehabilitation following TKA surgery was equally as effective as conventional treatment during a specific rehabilitation period [55]. Christiansen et al. [56] performed a randomized controlled trial with blinded evaluators. The authors examined the effects of weight-bearing (WB) biofeedback training on WB symmetry and functional joint moments following unilateral total knee arthroplasty. They used the Nintendo Wii Fit Plus game associated with the Wii Balance Board (Nintendo of America Inc, Redmond, Washington, DC, USA). The addition of a six-week intervention of WB biofeedback training to standard rehabilitation post-TKA did not improve functional WB symmetry. The feasibility and safety of the Nintendo Wii game console were also evaluated by Ficklscherer et al. [57]. A prospective, randomized, controlled study was conducted comparing standard physiotherapy vs. physiotherapy plus game console training in patients with ACL repair or TKA. The authors demonstrated that physiotherapy using the Nintendo Wii gaming console does not negatively influence the outcome and see an opportunity whereby additional training with a gaming console for an extended time could lead to even better results. Chung-Ho Su [58] developed a Kinect-based rehabilitation system using a Microsoft Kinect™ platform (a motion-sensing input device produced by Microsoft for the Xbox 360 video game console and Windows PCs) to assist the rehabilitating TKR patients. The author performed a study with a quasi-experimental design. The experimental group showed significant improvement in knee flexion. Chung-Ho Su concluded that motivation influences the effectiveness of rehabilitation and that the Kinect-based rehabilitation system acceptance by rehabilitates was indeed able to strengthen their motivation and improve their rehabilitation outcomes [58]. Roig-Casasús et al. [59] evaluated the influence of specific balance-targeted training using a dynamometric platform (Balance System™ SD, Biodex Medical Systems Inc, Shirley, New York, NY, USA) on the overall state of balance in patients undergoing TKA. The authors carried out a prospective, randomized 2-arm clinical trial. They demonstrated that a 4-week functional training protocol that included a dynamometric platform in the training methods enhanced the balance performance of older adults in the early TKR postoperative stage to a greater extent than a traditional functional training protocol [59].

Recently, Berton et al. [60] described the state of the art in VR, AR, gamification, and telerehabilitation for orthopaedic rehabilitation. The authors underlined the need for future research to demonstrate the advantages of these technologies compared to face-to-face orthopaedic rehabilitation. Lingfeng Li [61] randomly divided 88 patients who underwent knee joint injury surgery into an experimental group (treated with AR-based rehabilitation after the surgery) and a control group (treated with traditional rehabilitation). Li stated that, compared with conventional methods, the rehabilitation training method based on AR showed substantial advantages in alleviating postoperative pain and helping structural and functional recovery [61].

Another area where applications of immersive technologies are being developed is gait analysis. A conventional optical-based gait analysis laboratory employs expensive stereophotogrammetric motion capture systems. Several studies [62,63,64] proposed the application of VR/AR devices to carry out gait analysis, significantly reducing the cost, with similar performance to conventional technologies.

## 4. Present and Future Directions

Due to its technical characteristics with simple computer developments, the optical technological expansion achieved, and the economic sustainability it allows, the combination of three-dimensional planning on virtual models and the visual representation in AR of these models in real-time in different knee surgery techniques using fiducial markers from the ArUco library can be beneficial in the different aspects that we previously mentioned as desirable: increased precision to traditional surgical techniques, increased surgeon’s attention in the surgical field, reduction of possible radiation exposure, reduction of procedure time and costs, improvement of operating room efficiency, teaching benefit and the creation of positive synergies between engineering and medicine that favour the development of bioengineering. All these aspects are on the way to the pursuit of surgical excellence, which must positively impact outcomes and improve patients’ health-related quality of life. We can see the result of the axial correction of a tibial osteotomy in all three planes before performing it. We can project the trajectories of multi-ligament lesion tunnels in the knee before drilling into the bone. We can know how deep an osteotome penetrates a femur’s metaphysis without using the X-ray image intensifier. Today, it is possible.

It is a technological innovation incorporated into surgical activity and is present in our knee surgeries. There are still several aspects to demonstrate, such as its usefulness, accuracy, efficiency, and economic sustainability. We are working on different aspects related to AR, such as the accuracy of ArUco markers in vitro. In this modernity in which we move in recent robotic knee surgery, with extended realities in that continuum between virtuality and reality [25] and artificial intelligence, everything will make sense if it finally contributes to augmented humanity, as defined by Guerrero et al. [65]: “Augmented humanity is a human–computer integration technology that proposes to improve capacity and productivity by changing or increasing the normal ranges of human function, through the restoration or extension of human physical, intellectual and social capabilities”.

## Figures and Tables

**Figure 1 jpm-13-00727-f001:**
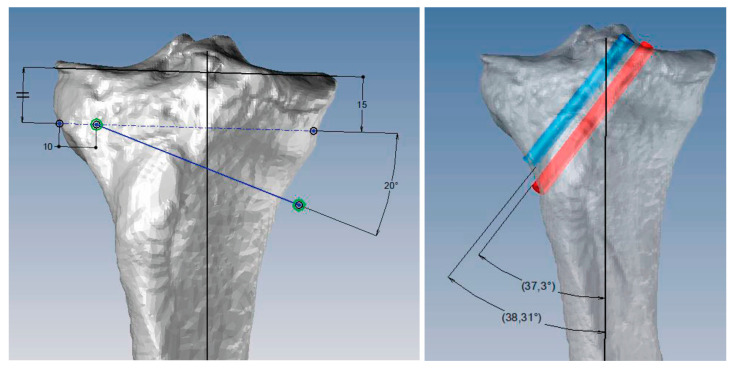
Images of the virtual model with the planning of the tibial osteotomy and the tunnels for the anterior and posterior anchorage of the medial meniscal transplant.

**Figure 2 jpm-13-00727-f002:**
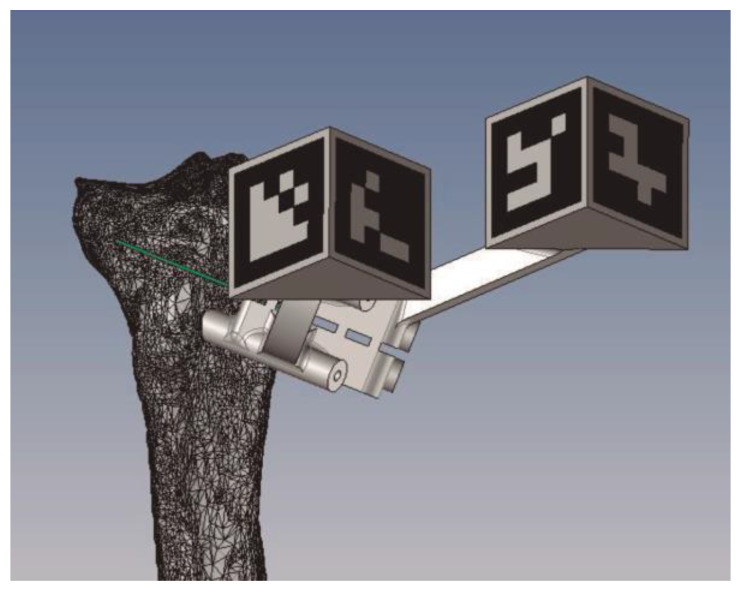
Detail of the jig design to perform the osteotomy with the ArUco markers.

**Figure 3 jpm-13-00727-f003:**
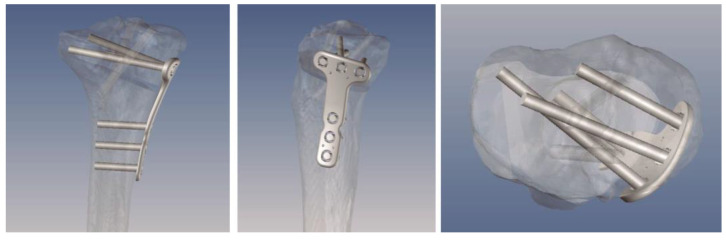
Virtual modelling of the plate and screw placement avoiding the tunnel trajectory for anchoring the medial meniscal transplant.

**Figure 4 jpm-13-00727-f004:**
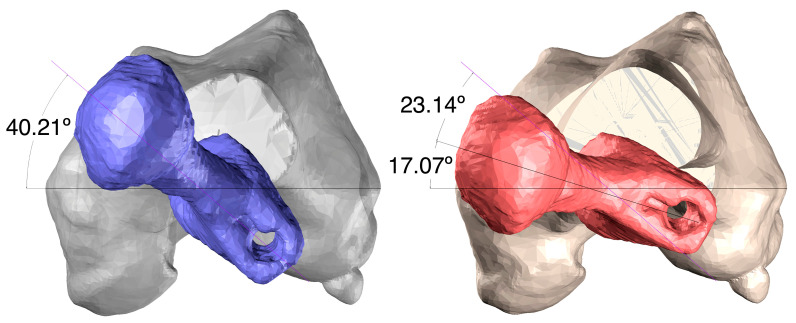
Details of the analysis of the rotational distortion on the virtual model and of the proposed corrections.

**Figure 5 jpm-13-00727-f005:**
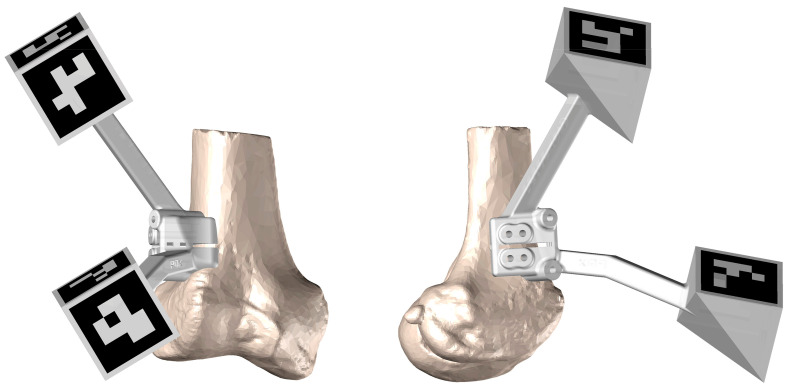
Cutting jig with ArUco markers for guiding femoral osteotomy under AR control.

## Data Availability

Not applicable.

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
