# Peer review of "Integration of Square Fiducial Markers in Patient-Specific Instrumentation and Their Applicability in Knee Surgery"

_jpm, 2023, doi:10.3390/jpm13050727_

Round 1

Reviewer 1 Report

Discussion in the "narrative review" should be performed more "narrative", and not as a list of the literature, otherwise a "systematic" review following the PRISMA guidelines could be more accurate.

Conclusion: It is still a scientific work, conclude pointing out the results of the AR in a scientific manner; delete lines 387-388.

Author Response

Dear Reviewer 1,

First, we would like to thank you for your comments and for allowing us to address the issues you raise to improve the manuscript’s quality. We have modified those aspects that you have indicated to us.

I will provide a point-by-point response to your comments and objections and indicate the modifications I propose to the manuscript to incorporate your comments.

Discussion in the "narrative review" should be performed more "narrative", and not as a list of the literature, otherwise a "systematic" review following the PRISMA guidelines could be more accurate.

We have tried to improve the narrative aspect of the description of the existing literature. This manuscript does not aim to review the literature systematically; therefore, we have not applied the PRISMA criteria. We have ensured that the narrative review section facilitates the understanding of the subject, describing it broadly with a theoretical foundation and trying to make it possible to contextualise, problematise and visualise proposals and new perspectives.

Conclusion: It is still a scientific work, conclude pointing out the results of the AR in a scientific manner; delete lines 387-388.

We have deleted the sentences you mention. The following paragraphs can summarize in a scientific and, at the same time, humanistic or philosophical way the meaning of AR and all the technologies we are incorporating into our daily practice. Innovation will make sense if it contributes to an increased humanity, i.e., to compensate for our limitations to improve our diagnostic and treatment capacity, compensating for the possible inaccuracy that we as humans may present.

Finally, we have modified by mechanical and constructive paraphrasing and by increasing the bibliographic citation calls (which, in some cases, we have previously omitted to avoid self-citation) the publisher’s information of a high similarity index.

Reviewer 2 Report

The authors describe the integration of fiducial markers in planning knee surgeries and to perform a narrative description of the latest publications on AR applications in knee surgery. The paper is interesting and well writen, with a few minor errors found and listed in sequence.

Why was marker-based AR chosen instead of a markerless-based AR solution (for instance, "A Novel Visualization System of Using Augmented Reality in Knee Replacement Surgery: Enhanced Bidirectional Maximum CorrentropyAlgorithm") ? Plase justify this in the text.

"AR projection-based" -> "projection-based AR"

"performs the magnitude operation (a-b); " - > the formula immediately following this text is not clear and does not represent values from a and b. a more adequate formula would be sqrt((ax-bx)^2 + (ay-by)^2 + (az-bz)^2).

"Segmentation requires a trained operator, as no fully automated segmentation algorithms exist [6,30]." -> This is not true. There are plenty solutions of automatic segmentations. One specifically related to knee segmentation is "Study on the accuracy of automatic segmentation of knee CT images based on deep learning" (https://pubmed.ncbi.nlm.nih.gov/35570625/)

Please highlight how the proposed project advances the state of the art. Please compare the proposed solution with state of the art ones, clearly showing how existing solutions work anf how They surpass the state of the art.

Author Response

Dear Reviewer 2,

First, we would like to thank you for your comments and for allowing us to address the issues you raise to improve the manuscript’s quality. We have modified those aspects that you have indicated to us.

We also want to thank you for finding the manuscript interesting and well-written. It is an encouragement and an important recognition for us.

I will provide a point-by-point response to your comments and objections and indicate the modifications I propose to the manuscript to incorporate your comments.

The authors describe the integration of fiducial markers in planning knee surgeries and to perform a narrative description of the latest publications on AR applications in knee surgery. The paper is interesting and well writen, with a few minor errors found and listed in sequence.

Why was marker-based AR chosen instead of a markerless-based AR solution (for instance, “A Novel Visualization System of Using Augmented Reality in Knee Replacement Surgery: Enhanced Bidirectional Maximum Correntropy Algorithm")? Please justify this in the text.

Augmented Reality does not provide a basis in the three axes of space for precise measurements, i.e., quantitatively speaking, we cannot measure or take a dimension from point A (x, y, z) to point B (x, y, z).

We have added to the manuscript: "The planning and navigation system we have used combines virtual and AR. Planning and calculations are performed on the virtual model. The markers are needed as references to project the virtual planning onto AR. AR in isolation does not provide a basis in the three axes of space for precise measurements from point A (x, y, z) to point B (x, y, z)."

"AR projection-based" -> "projection-based AR"

It was changed in line 89. Thank you.

"performs the magnitude operation (a-b); " - > the formula immediately following this text is not clear and does not represent values from a and b. a more adequate formula would be sqrt((ax-bx)^2 + (ay-by)^2 + (az-bz)^2).

We have changed the formula, as you propose. You are right; it better represents the operation (a-b) in Cartesian space.

“Segmentation requires a trained operator, as no fully automated segmentation algorithms exist [6,30].” -> This is not true. There are plenty solutions of automatic segmentations. One specifically related to knee segmentation is “Study on the accuracy of automatic segmentation of knee CT images based on deep learning” (https://pubmed.ncbi.nlm.nih.gov/35570625/)

Thank you for your comment. Unfortunately, I could only read the entire article (DOI 10.7507/1002-1892.202201072) after filtering it through an electronic translation system (DeepL), as I do not know Chinese. I used the same system to read the translation of article DOI 10.7507/1002-1892.202212008. The literature I had read and my experience of more than ten years planning surgeries on 3D virtual models and using PSI in TKA surgery made me aware of how dependent operator segmentation is. The advance of IA is indeed unstoppable, and it is only a matter of time before neural network models surpass the accuracy of the human operator, so I have changed the paragraph to show authors who argue for operator dependence and those who argue for reliable automation in segmentation. The rewritten text reads: “The bioengineers obtain the 3D model of each specific knee to be operated on with its anatomical characteristics and particularities (in the case of CT studies of the bone morphology of the knee) using segmentation. According to some authors, segmentation requires a trained operator, as no fully automated segmentation algorithms exist [6,30]. Other authors, on the other hand, have published remarkable accuracy of automatic segmentation by implementing a neural network architecture based on deep learning [31–33]. In our usual practice, this process is carried out by the bioengineers using Mimics® software (Materialise, Leuven, Belgium) for the segmentation process. Mimics® is an interactive tool for the visualization and segmentation of CT and Magnetic Resonance Imaging (MRI) images and 3D rendering of objects.”

Please highlight how the proposed project advances the state of the art. Please compare the proposed solution with state-of-the-art ones, clearly showing how existing solutions work and how They surpass the state of the art.

We have tried to respond to your request by adding the following paragraph:

“Due to its technical characteristics with simple computer developments, the optical technological expansion achieved, and the economic sustainability it allows, the combination of three-dimensional planning on virtual models and the visual representation in AR of these models in real-time in different knee surgery techniques using fiducial markers from the ArUco library can be beneficial in the different aspects that we previously mentioned as desirable: increased precision to traditional surgical techniques, increased surgeon’s attention in the surgical field, reduction of possible radiation exposure, reduction of procedure time and costs, improvement of operating room efficiency, teaching benefit and the creation of positive synergies between engineering and medicine that favour the development of bioengineering. All these aspects are on the way to the pursuit of surgical excellence, which must positively impact outcomes and improve patients’ health-related quality of life.”

Finally, we have modified by mechanical and constructive paraphrasing and by increasing the bibliographic citation calls (which, in some cases, we have previously omitted to avoid self-citation) the publisher’s information of a high similarity index.